# Killer Knots: Molecular Evolution of Inhibitor Cystine Knot Toxins in Wandering Spiders (Araneae: Ctenidae)

**DOI:** 10.3390/toxins15020112

**Published:** 2023-01-28

**Authors:** Michael S. Brewer, T. Jeffrey Cole

**Affiliations:** Department of Biology, East Carolina University, Greenville, NC 27858, USA

**Keywords:** spider, selection, transcriptome, proteome, inhibitory cystine knot, molecular evolution, phylogeny

## Abstract

Venom expressed by the nearly 50,000 species of spiders on Earth largely remains an untapped reservoir of a diverse array of biomolecules with potential for pharmacological and agricultural applications. A large fraction of the noxious components of spider venoms are a functionally diverse family of structurally related polypeptides with an inhibitor cystine knot (ICK) motif. The cysteine-rich nature of these toxins makes structural elucidation difficult, and most studies have focused on venom components from the small handful of medically relevant spider species such as the highly aggressive Brazilian wandering spider *Phoneutria nigriventer*. To alleviate difficulties associated with the study of ICK toxins in spiders, we devised a comprehensive approach to explore the evolutionary patterns that have shaped ICK functional diversification using venom gland transcriptomes and proteomes from phylogenetically distinct lineages of wandering spiders and their close relatives. We identified 626 unique ICK toxins belonging to seven topological elaborations. Phylogenetic tests of episodic diversification revealed distinct regions between cysteine residues that demonstrated differential evidence of positive or negative selection, which may have structural implications towards the specificity and efficacy of these toxins. Increased taxon sampling and whole genome sequencing will provide invaluable insights to further understand the evolutionary processes that have given rise to this diverse class of toxins.

## 1. Introduction

Animals of numerous phyla have independently evolved venom to inject into and cause harm to other animals for the purposes of either subduing prey, fending off predators, or competing with rival mates [1,2,3]. Venomous animals constitute ~15% of all described animal biodiversity, with spiders representing the largest group of venomous animals with approximately ~50,000 species currently described [4,5]. Recent phylogenetic investigations have revealed that spiders recruited inhibitor cystine knot (ICK) toxins into their venom arsenal via a duplication event and subsequent neofunctionalization ~300 million years ago [6]. Several subsequent rounds of duplication events and subfunctionalization throughout their evolutionary history has given rise to a vast library of toxins that has allowed spiders to succeed as generalist predators that can incapacitate a broad array of prey items by expressing hundreds to thousands of unique venom components spanning an estimated million unique pharmacologically active components spread across all species of spiders [7,8,9].

The exploration of venom components in spiders has largely focused on the small handful of spiders with dangerous bites to humans, which has greatly limited and biased the current understanding and biological context of the diverse array of toxins expressed by spiders. A prime example is the medically relevant Brazilian wandering spider *Phoneutria nigriventer*, which has been the focus of numerous investigations to delineate active noxious components [10,11]. One such component is toxin Tx2-6 which is the responsible agent for priapism, occasionally an envenomation symptom of *Phoneutria* in human males [12]. The most notable component of their venom is toxin Tx1, which exerts inhibitory effects on neuronal sodium channels in a highly selective manner and has a lethal median dosage (LD_50_) of 47 μg/kg in *Mus musculus* [13]. This is nearly four times more toxic than the lethal nerve agent Sarin, which is classified as a Schedule 1 substance by the Chemical Weapons Convention of 1993 and has an LD_50_ of 172 μg/kg in *M. musculus* when injected subcutaneously [14,15].

ICK toxins make up the majority of the venom composition in spiders [16]. The core ICK cysteine framework consists of three pairs of cysteines forming disulfide bridges between $C_{1}$-$C_{4}$, $C_{2}$-$C_{5}$, $C_{3}$-$C_{6}$ to take on an unusually stable conformation [17,18]. A recent investigation of the venom gland transcriptome and proteome of *P. nigriventer* (Keyserling, 1891) revealed that their noxious venom components belong to a diverse class of ICK toxins [19]. That same investigation recovered 98 cysteine rich toxins that represented nine additional cysteine frameworks, six of which were verified to be ICKs. The number of cysteine residues per group ranged from six to fourteen. They also spanned a broad range of predicted functionality, from ion channel modulators of varying specificity (Ca${}^{+2}$, K${}^{+}$, and Na${}^{+}$), to protease inhibitors and NMDA receptor modulators. The cysteine-rich peptide toxins represented 93.24% of the relative abundance of peptides expressed in the venom when accounting for expression levels.

The discovery of a vast library of ICK toxins expressed by *P. nigriventer* has led to numerous questions about their evolutionary origins. In this study, we used venom gland transcriptomics and proteomics with increased taxon sampling from phylogenetically distinct lineages to provide the first comprehensive framework to test hypotheses about the molecular evolution of ICK toxins in wandering spiders.

## 2. Results

### 2.1. Venom Gland Transcriptome and Proteome

The 48 assemblies in this analysis included an average transcript recovery of 106,410 (s.d = 43,708), representing an average of 84,474 (s.d = 32,908) genes as designated by Trinity (Table 1). The 21,936 genes with alternative transcripts designated by Trinity had an average of 3.04 isoforms (s.d = 1.99). From the longest isoforms, we recovered on average 12,173 complete coding sequences per species (s.d = 4993), with an average amino acid length of 264 (s.d = 9).

### 2.2. Phylogenetic Results

The average percentage of complete BUSCO hits within the assemblies was 84.96% (s.d = 15.48%), and an average of 60.33% (s.d = 8.90%) were single copy. 245 BUSCO loci met the threshold of 60% of species represented with at least 50% of the sequences being nonidentical. The untrimmed alignments had an average matrix width of 348.8 nucleotides (s.d = 179.0); trimming the alignments reduced the average to 314.7 nucleotides (s.d = 162.8). The total size of the concatenated matrix was 78,594 nucleotides.

There were no topological differences between the concatenated matrix phylogeny and the ASTRAL species tree. The only topological difference that occurred between this study and Cheng et al. 2018 [20] was that we recovered Oxyopidae as sister to a clade comprising Ctenidae + Psechridae + Lycosidae + Pisauridae, instead of being sister to the Thomisidae. Temperate zone North American ctenids do not form a single lineage, as *Anahita punctulata* is sister to a clade comprising the North American *Ctenus*-*Leptoctenus* lineage plus the Neotropical *Phoneutria*-*Isoctenus* clade. Within the Ctenidae, the genus *Ctenus* is polyphyletic with respect to *Ctenus corniger*, the only Old World representative of the family, being sister to all New World member ctenids included here (Figure 1). This recapitulates previous findings that *Ctenus* has served as a repository for taxonomically problematic species [21,22,23,24].

### 2.3. Inhibitor Cystine Knot Annotation

A total of 1259 cysteine rich peptides were recovered that met the following criteria: complete coding sequence (with start and stop codon) signal peptide present, less than 200 amino acids, mature peptide had at least 6 cysteines, and was a match with the KNOTTIN database from either a BLAST search or HMMER with an e-value cutoff of $1\times 10^{-3}$. On average each sample contained 26.6 cysteine rich peptides (s.d = 8.3). Cysteine rich peptides made up less than 10% of the total peptides in the proteomes of *C. exlineae* and *C. hibernalis*, though in at least two samples they made up over 50% of the relative abundance when accounting for expression level (Table 2).

SiLiX grouped the cysteine rich peptides into 53 putative gene families. The largest family comprised 1148 peptides, and the top seven frameworks corresponding to the ICKs described by [19] represented 960 putative ICKs (Table 3). The largest cysteine framework recovered was C8.0 with 538 peptides, whereas C6.0, the next largest, represented 123 peptides (Figure 2).

The largest cysteine framework (C8.0), was the most abundant framework for all species except for *Homalonychus theologus*. One framework (C12.1) was only recovered in Psechridae and Ctenidae. A novel cysteine framework (C10.1) not reported by Diniz et al. [19], was not recovered in *P. nigriventer*, though it was recovered in five other species of Ctenidae (Table 4).

### 2.4. Disulfide Connectivity Predictions

Disulfide connectivity predictions varied greatly between the different prediction approaches; predictions for peptides with fewer cysteines were more consistent between approaches (Figure 3). The three disulfide bridges homologous to all cysteine frameworks were cross referenced to peptides with the same cysteine framework in Arachnoserver and used to guide the subsequent multiple sequence alignment (Figure 2).

### 2.5. Phylogenetic Tests for Selection

The first four inner cysteine loops shared by all ICKs were aligned following the schema defined in Table 5. This resulted in a multiple sequence alignment with a width of 80 amino acids for 626 peptides with no redundant coding sequences and no sequences that failed the Chi-squared sequence composition test.

Based on the reconstructed phylogeny of included ctenids (Figure 2) all other cysteine frameworks appear to have originated from framework C8.0. Framework C6.0 appears to have evolved via a loss of a pair of cysteines ($C_{6}$ and $C_{7}$) from the original C8.0 framework. The largest monophyletic grouping of C6.0 was entirely unique to ctenids. The framework C10.1 represents an entirely separate lineage and bridging pattern from C10.0 and is monophyletic. The remaining frameworks are mostly monophyltic and have evolved in a step-wise fashion from the ancestral C8.0 framework, with several examples of convergent evolution and reversals within some clades.

BUSTED, with synonymous rate variation found evidence (LRT, *p*-value ≤ 0.05) of gene-wide episodic diversifying selection across the entire gene phylogeny (Table 6). Therefore, there is evidence that at least one site on at least one branch has experienced diversifying selection. The site by site variation in test statistics is visualized in Figure 4, though BUSTED does not possess the statistical power to infer which specific sites or branches display evidence of episodic diversifying selection.

FUBAR did not find evidence of pervasive positive/diversifying selection at any sites, but evidence of negative/purifying selection was detected at 57 sites with a posterior probability of 0.9. The line of best-fit from a linear regression of $d_{S}$ as the independent variable and $d_{N}$ as the dependent variable for each of the 80 sites had a slope of 0.49 ($F_{1,78}=44.78,R^{2}=0.36,p=3.02\times 10^{-9}$). Only four sites had $d_{N}$ estimates that exceeded $d_{S}$, though the highest posterior probability of those sites was 0.54 (Figure 5).

MEME found evidence of positive/diversifying selection under a portion of gene phylogeny branches at 12 sites with *p*-value threshold of 0.05, after correcting for multiple testing (Figure 6). Four were within the first loop between the first and second cysteine residues. None were within the second loop between the second and third cysteine residues. Two of those sites were directly upstream of the adjacent pair of cysteines (sites 26 and 27 of the alignment).

aBSREL found evidence of episodic diversifying selection on two out of 1158 branches in the gene phylogeny. Significance was assessed using the Likelihood Ratio Test at a threshold of *p* ≤ 0.05, after correcting for multiple testing. One branch was a clade of four peptides with the C8.0 cysteine framework expressed by one ctenid (*C. corniger*), two oxyopids (*Peucetia longipalpis* and *Oxyopes* sp.) and one psechrid (*Fecenia protensa*). The other branch was a clade of 19 peptides with the C14.0 cysteine framework which comprises only New World ctenids (i.e., excluding *C. corniger*).

BGM found 30 pairs of coevolving sites (posterior probability ≥ 0.95), of which 10 had a posterior probability ≥ 0.99. Of the 30 pairs of coevolving sites, 11 were at least three residues apart, whereas the furthest distance between two coevolving sites was 66 amino acid residues. Of particular interest was the fifth amino acid residue downstream of the first cysteine, which was found to be coevolving with three other residues, more than any of the others. Site 5, was found to be coevolving with sites 9, 23 and 35. Sites 5 and 9 are both found within the first loop between cysteines one and two, whereas site 23 is in the middle of the second loop, and site 35 is centrally located in the third loop.

## 3. Discussion

In this study, we identified and characterized the molecular evolution of 626 unique coding sequences for ICK peptides in wandering spiders and their free-hunting lycosoid relatives. The molecular functionality of these toxins is still unknown for the most part. Increased efforts in neurophysiology assays and molecular modeling will allow broader insights into the evolution of the molecular targets of these toxins. The best disulfide connectivity servers currently available were incredibly imprecise at predicting disulfide connections in spider ICKs [25,26,27]. This is especially true for ICK structural elaborations with more than four cysteine pairs, illustrating the need for a spider-specific approach to elucidating structural predictions. Unfortunately, a large bottleneck in making that a possibility is empirical investigations in determining the structure of ICK elaborations in spiders.

It appears that the original ICK toxin in spiders may not be what is typically referred to as the “core” ICK cysteine framework with three pairs of cysteines. Instead, the most abundant cysteine framework with four pairs appears to be the original ICK toxin, and the 6-cysteine framework evolved from the 8-cysteine framework via a loss of a pair of cysteines. Though this study only focused on lycosoid spiders, 8-cysteine toxins appear to be the most abundant throughout the spider tree of life, so a more comprehensive analysis of ICKs across all spiders may yield similar results.

It is postulated that antagonistic co-evolution through predator-prey interactions has shaped venom function via reciprocal selective pressures in an evolutionary “arms race” [28,29,30,31]. At the protein level, selective pressures on venom have been observed through patterns of rapid evolution of amino acid sequences [32]. More specifically, according to the Rapid Accumulation of Variations in Exposed Residues (RAVER) model of venom evolution, structurally important residues receive strong negative selection while there is a rapid accumulation of variation in the molecular surface of the toxin under a coevolutionary “arms race” scenario [33]. The coevolution of venom resistance in prey and increasingly potent venom in the predator are theorized to exert reciprocal selection pressures [33].

Broadly speaking, there was evidence of gene-wide episodic diversifying selection in ICK toxins of lycosoid spiders. There was no evidence of pervasive positive selection in any of the codon sites of the ICK alignment. This is consistent with what has been reported in previous tests of pervasive positive selection in spider ICKs [2]. ICKs in spiders date back ~300 MY, so it is not unusual that ~70% of the amino acid sites demonstrated evidence of negative selection, because evolution “erases its traces” of early bouts of positive selection with persistent negative selection to preserve the potency of the toxin [2,34]. What is particularly striking, though, is that evidence of episodic positive selection was detected in a portion of branches for 12 amino acid sites. This is consistent with the two-speed model of venom evolution proposed by Sunagar and Moran 2015 [2], in which positive selection pervades early in venom evolution (such as what is observed in the toxins of contemporary snakes and cone snails) followed by bouts of negative selection, then subsequent bouts of positive selection. These later episodic bouts of positive selection may be indicative of ecological specialization, such as dietary shifts and range expansions, resulting in a rapid diversification of venom arsenal.

Structurally, none of the residues between the first and second cysteine showed evidence of episodic diversification. This could indicate that those residues are necessary to maintain structural integrity and sustain venom potency. One of the two branches on the ICK phylogeny that had evidence of positive selection was a clade of 19 14-cysteine ICKs entirely unique to ctenids. It is possible that this ICK elaboration has played an important role in the range and diet expansion of ctenids. There was also strong evidence of amino acid co-evolution between one residue within the first loop and another amino acid four residues upstream in the same loop and two additional separate residues found midway through the second and third loop. This may indicate that these residues play an important role in the structural integrity or potency of the venom as they had a much higher than expected rate of co-occurrence.

## 4. Conclusions

In this study, we provided evolutionary insights into the ICK toxins of spiders. These insights may prove useful in the field of bioprospecting and peptide design, in which the ICK scaffold is useful for agricultural and pharmacological applications. What remains unresolved are the evolutionary mechanisms giving rise to molecular functions of these toxins, which will become a possibility as more structural and functional assays in spider ICKs are performed. None of the species included in our analysis have publicly available genome sequences, so our analyses relied on incomplete transcriptomic and proteomic data. However, we demonstrated that these toxins exist as multi-copy gene families across different species. What has yet to be determined are the specific mechanisms that have given rise to these large gene families. Sequencing the genomes of these spiders would provide valuable insights into the evolution of ICK toxins in spiders and finally allow investigations regarding the diversification and formation of cysteine framework elaborations of ICKs in spiders.

## 5. Materials and Methods

### 5.1. Taxon Sampling

Three adult males and females were collected from all members of Ctenidae (Arthropoda: Araneae) in the United States (taxonomically identifed by TJC), excluding the narrow Texas cave endemic *Ctenus valveriensis* Peck, 1981. Specimens of *Ctenus hibernalis* Hentz, 1844, *Ctenus exlineae* Peck, 1981, *Ctenus captiosus* Gertsch, 1935, *Leptoctenus byrrhus* Simon, 1888, and *Anahita punctula* (Hentz, 1844) were collected from the following respective localities: North-Central Alabama (33.462, −86.788), Northwest Arkansas (34.376, −94.029), Central Florida (29.0823, −81.578), South-Central Texas (29.831, −99.573), and Northwest Georgia (34.049, −85.381). Four male and female *Phoneutria nigriventer* (Keysterling, 1891) along with four female *Isoctenus* sp. supplied by and taxonomically identified by Antonio Brescovit. Ctenidae belongs to the superfamily Lycosoidea, which is a member of a clade of spiders that possess a retrolateral tibial apophysis (RTA), a backward-facing projection on the tibia of the male pedipalp. To allow for an investigation of the broader evolutionary context of ICK toxins in wandering spiders, whole body transcriptomes from outgroups within the RTA clade were retrieved from NCBI’s Short Read Archive.

### 5.2. Venom and RNA Isolation

Spiders were housed in a 500 cm^3^ plastic container and were watered, cleaned and fed crickets (*Acheta domesticus*) weekly. The temperature was maintained between 22–25 °C, which is near the average temperature that these animals experience in their natural habitats. Prior to venom collection, individuals were anesthetized with CO${}_{2}$ using a modified procedure as described by [35]. Venom was collected using electrostimulation with 7 V of AC current, similar to previous studies [36,37,38]. Anesthetized individuals were placed on clamped forceps attached to an electrode. One prong of the forceps was wrapped in non-conductive insulating tape to create a point of contact for the spider that retards current, while the other prong of the forceps was wrapped with a cotton thread and soaked in saline to create a point of contact with the spider to promote electrical conductivity. A capillary tube was then placed over the fang in order to collect the venom. Finally, the second electrode was placed on a syringe connected to a vacuum pump which was touched to the base of the chelicerae in order to complete the circuit and allow the muscles around the venom gland to contract and eject venom into the capillary tube while simultaneously allowing regurgitate to be vacuum pumped through the syringe to prevent contamination. The collected venom was then stored at −80 °C. Two days after venom milking, the venom glands of each ctenid were dissected out, whole RNA was isolated from the venom glands from the five U.S. species and the two Brazilian species using TRIzol^®^ (Life Technologies, Carlsbad, CA, USA) and the Qiagen RNeasy kit (Qiagen, Valencia, CA, USA). RNA concentration and integrity was evaluated using Quant-iT™ PicoGreen and Bioanalyzer.

### 5.3. Sequencing and Processing

The RNA extractions were sent to the Genomic Services Lab at HudsonAlpha (Huntsville, AL, USA) for library preparation with poly(A) selection and sequencing on a 100 bp paired-end run on an Illumina HiSeq 2500, comprising 25 million reads forward and reverse (50 million total reads) per barcoded sample on a single lane. For all samples sequenced after 2018 (Briazillian samples and *C. captiosis*), due to sequence facility updates, the same setup was used but with a single Illumina NovaSeq lane. Additionally, RNAseq reads from all outgroup species were retrieved from NCBI Short Read Archive.

Prior to assembly, FASTP v 0.19.6 [39] was used to remove adapters, correct sequencing errors, and trim low-quality base calls, ensuring maximal accuracy of transcript recovery [40]. De novo assemblies typically recover an unexpectedly large number of transcripts, sometimes well over 100,000 [41]. This happens for three main reasons. First, an increased depth of sequencing combined with improved transcript recovery algorithms increases the recovery of transcripts that are expressed at levels lower than would otherwise be considered biologically relevant. Second, common contaminants, such as bacteria and fungi, unavoidably make their way into samples, thus inflating the mRNA transcript pool diversity. Third, in eukaryotic systems, alternative splicing yields a significant increase in recoverable transcripts per gene locus as isoforms. Further, when several RNA-seq experiments from different species are sequenced together, cross contamination inevitably occurs [42,43,44,45,46,47,48]. The aforementioned issues can have tremendous effects on downstream phylogenetic inferences made from problematic transcripts [49]. To alleviate these issues, reads were first mapped to a transcript database of common contaminants (bacteria, fungi, human, and nematodes) using salmon v1.3.0 with default mapping parameters, all unmapped reads were retained as the finalized library of processed reads [50].

### 5.4. Transcript Reconstruction and Expression Quantification

The processed reads were de novo assembled into transcripts using TRINITY v2.8.4 [51]. An important aspect of elucidating the functional relevance of a given protein is the quantification of its expression level. This can be achieved at the transcript level through quantifying read coverage by mapping the reads from each sample back to their assembled transcripts. Salmon uses a quasi-mapping approach and is one of the fastest, most efficient, and most accurate methods for quantifying expression in RNAseq experiments [50]. Since TRINITY also assembles splice variants and alleles of the same gene, these were consolidated into SuperTranscripts [52] prior to mapping so that the inferred expression values are at the gene level and not the isoform level. SuperTranscripts are formed by collapsing common and unique regions of sequences among splicing isoforms into a singular consolidated linear sequence. Then, the SuperTranscripts were used as mapping targets, and the processed reads were pseudomapped using salmon to quantify expression levels as Transcripts per million (TPM).

### 5.5. Venom Proteomics

Venom from *C. hibernailis* and *C. exlineae* were pooled for proteomic characterization. To characterize the venom profiles, venom proteins were isolated using HPLC followed by MALDI-TOF Mass Spectrometry. MALDI-TOF is the preferred method for mass analysis in proteins due to only a single charge applied to analytes, compared to commonly used ESI techniques which apply multiple charges to analytes and complicate downstream analysis. The finalized dataset of candidate venom encoding transcripts were elucidated by cross-referencing proteomic MS data to the transcriptome using the CRUX pipeline [53].

### 5.6. Locus Sampling

Coding sequences within transcripts were inferred using TRANSDECODER v3.0.1 [54]. TRANSDECODER uses the following criteria to identify the single best coding sequence in a given transcript: available open reading frame (ORF) of a minimum length of 30 codons, log-likelihood score of the coding sequence, predictions of start and stop codons as refined by a Position-Specific Scoring Matrix. The completeness of the assemblies was evaluated using BUSCO v3.0.2 (Benchmarking Universal Single-Copy Orthologs) [55] with the arthropod database. In addition to providing a metric of assembly completeness, complete BUSCO transcripts serve as a robust set of loci for phylogenetic analysis.

### 5.7. Phylogenetic Reconstruction

A well supported phylogeny provides a necessary evolutionary framework for comparative analysis of venom evolution. To reconstruct the North American ctenid species relationships, additional RNAseq reads from 20 outgroup species were retrieved from NCBI Short Read Archive. Loci sampling for phylogenetic analysis involved the following procedure. Only complete coding sequences (begins with start and ends with stop codon) inferred from TRANSDECODER that were the longest isoform of a given TRINITY gene assignment were used for this analysis. Coding sequences that contained a single complete match to a BUSCO term were retrieved from the assemblies. Multiple protein alignments were generated with MAFFT v7.221 [56] for BUSCO matches and retained if 30 out of 48 samples were present and <50% of the sequences were identical. This resulted in 245 BUSCO term alignments that were then trimmed with TRIMAL v1.4.1 [57] to remove uninformative sites using the -automated1 parameter to heuristically select the trimming method. Model selection was performed for the trimmed concatenated matrix to elucidate the best-fit model using the “TESTONLY” option of IQ-TREE v1.6.10 [58]. Subsequently, 1000 ultrafast bootstrap replicates were performed to calculate node support. Additionally, site and gene concordance factors were calculated as alternative support metrics. For the species tree analysis, the phylogenies of each gene were reconstructed using IQTREE with the same settings as previously mentioned [58]. The gene trees were then provided as input for ASTRAL v2.0 [59] to reconstruct the species phylogeny.

### 5.8. Inhibitor Cystine Knot Annotation

To identify inhibitor cystine knot toxins in the transcriptome assemblies, a database of verified ICKs from spiders was retrieved from the KNOTTIN database, which is a curated database of proteins with a disulfide through disulfide knot [60]. Only ICKs with complete coding sequences and verified disulfide connectivity were retained in the final verified ICK database. The database was provided as input to BLASTp to search against the inferred protein sequences from TRANSDECODER [61]. Additionally, a multiple sequence alignment was generated from the verified ICK database to create a Hidden Markov Model (HMM) that could be searched against the genomic protein sequences using HMMER v3.3.1 with the default settings [62]. For both BLASTp and HMMER, only matches with at least six cysteines and up to 200 amino acids in length were kept for downstream analysis. Additionally, putative matches were only kept if they contained a signal peptide as indicated by signalP v5.0, which predicts the presence of signal peptide cleavage sites [63]. A homology network of the finalized peptides was generated using an all against all BLASTp search, and then provided as input to SiLiX v1.2.11 to group the peptides into putative gene families [64]. Cysteine frameworks were designated using the following approach. Cysteines that were directly adjacent to each other were designated as CC. Cysteines separated by one residue were denoted as CXC. Finally, cysteines separated by more than one residue were designated as C−C. Each framework was given a numeric code to represent the number of cysteines they contain along with a unique identifier. To ensure that only ICKs are included in the analysis, only cysteine frameworks representing the top 80% of peptides in the largest family were included for downstream disulfide connectivity predictions and phylogenetic analysis. A non-redundant dataset was then created to only include unique coding sequences.

### 5.9. Disulfide Connectivity Predictions

Disulfide connectivity is of great importance in understanding structural homology in ICK toxins when sequence similarity is greatly reduced. Determining disulfide connectivity normally requires empirical structural validation but can be reasonably predicted using computational predictions. The number of possible disulfide bonds explodes in a combinatorial fashion, to the point where exhaustively comparing disulfide connection possibilities in peptides with more than 6 cysteines is not computationally tractable [65]. To alleviate this, a number of heuristic approaches have been developed. For this dataset the following four approaches were used to generate disulfide connection predictions for a random representative mature peptide from the top cysteine frameworks.

1.DISULFIND collectively decides the bonding state assignment of the entire chain using a Support Vector Machine binary classifier followed by a refinement stage [25]. DISULFIND v1.1 was used to generate a total of three alternative disulfide connection predictions.2.CYSCON uses a hierarchical order reduction protocol to identify the most confident disulfide bonds and then evaluate what remains using Support Vector Regression [26]. CYSCON v2015.09.27 was used to generate a single disulfide prediction per ICK representative per unique cysteine framework.3.CRISP v1.0 not only predicts disulfide bonds, but also the entire structure of a cysteine rich peptide by searching a customized template database with cysteine-specific sequence alignment with three separate machine learning models to filter templates, rank models, and estimate model quality [27]. CRISP was used to generate five structural models for each ICK representative per unique cysteine framework.

A chord diagram was constructed for each cysteine framework to demonstrate the variability in disulfide connectivities for every prediction attempt of each of the approaches using the D3 JavaScript library [66]. A consensus disulfide connection prediction of all the approaches in conjunction with previously published disulfide connections, as found through Arachnoserver [67], were used to generate the finalized disulfide connectivity predictions for the three disulfide bridges homologous to all cysteine frameworks.

### 5.10. Phylogenetic Tests for Selection

Aligning ICKs, or any cysteine rich peptide, is difficult due to nonhomologous cysteines mistakenly being aligned. Thus, the finalized consensus disulfide connectivities were used to inform the alignment of ICKs using a similar approach to Pineda et al. 2020 [6]. Rather than align everything at once and then manually adjusting misaligned cysteines followed by realignment of regions between the two adjusted cysteines, only amino acids between cysteines participating in disulfide bonds common to all ICKs in the dataset were used for the alignment. Additionally, the regions between homologous cysteines were aligned separately while using the barcode “WWYHWYYHMM” to replace flanking cysteines to prevent inner cysteines from misaligning with flanking cysteines similar to the approach by Shafee et al. 2016 [68]. The alignment was then provided as input to IQTREE v1.5.5 [58] to test the amino acid composition using a Chi-squared test. Any sequences that failed the test were removed from the alignment, and the sub-regions were realigned following the previously described procedure. The resulting alignment was reverse translated to form a coding sequence alignment using PRANK [69].

The same outgroup as used by Pineda et al. 2020 [6] (disulfide-directed β-hairpin from the whip scorpion *Mastigoproctus giganteus*) was added to the protein alignment using MAFFT v7.455 [56]. The phylogenetic relationships of the ICKs in this alignment were reconstructed using IQTREE and the default settings.

Adaptive molecular evolution is typically inferred in coding sequences by comparing ratios of the rates of nonsynonymous substitution and synonymous substitution ($d_{N}/d_{S}$ or ω), where $d_{N}$ exceeding $d_{S}$ indicates positive selection, $d_{S}$ exceeding $d_{N}$ indicates negative selection, and $d_{N}/d_{S}$ approaching unity indicates neutral evolution. The HYPHY [70] implementation of Branch-Site Unrestricted Statistical Test (BUSTED) for Episodic Diversification was used to assess whether a gene has experienced positive selection at at least one site on at least one branch. To determine if ICKs have experienced positive selection, the codon multiple sequence alignment and phylogeny were provided as input to BUSTED using default parameters.

In ICKs, specific amino acid sites may play an important role in the structure-function (e.g., binding specificity) and adaptive evolution. To identify specific amino acid sites that have undergone pervasive positive selection, the HYPHY implementation of a Fast, Unconstrained Bayesian AppRoximation (FUBAR) was used with the codon multiple sequence alignment and phylogeny provided as input and default parameters.

There may only be specific episodes where certain amino acids receive strong bouts of positive selection. To determine if amino acid sites have undergone positive selection, the HYPHY implementation of a Mixed Effects Model of Evolution (MEME) [71] was used to determine if certain amino acid sites have undergone episodic positive selection. The codon multiple sequence alignment and phylogeny were provided as input to MEME with default parameters and the phylogeny set as the background.

To evaluate specific instances on a phylogeny where positive selection has occurred, branch-site models are typically implemented. Much like how MEME is unable to statistically specify the exact branches within a site undergoing episodic positive selection, branch-site models are only able to identify specific branches where a certain portion of sites have undergone positive selection. To accomplish this, the HYPHY implementation of adaptive Branch-Site Random Effects Likelihood (aBSREL) [72] was used with default parameters, and the codon alignments and phylogeny were provided as input.

Aside from evaluating signatures of positive selection through calculations of codon substitution rates, we also investigated the co-occurrence between amino acid positions in ICKs, which may provide useful inferences into the evolution of their structure/function. This can be achieved using the HYPHY implementation of the Bayesian Graphical Model (BGM) [73], which maps amino acid substitutions to a phylogeny and reconstructs ancestral states for a given model of codon substitution rates that is then followed up by a series of 2 × 2 contingency table analyses.

## Figures and Tables

**Figure 1 toxins-15-00112-f001:**
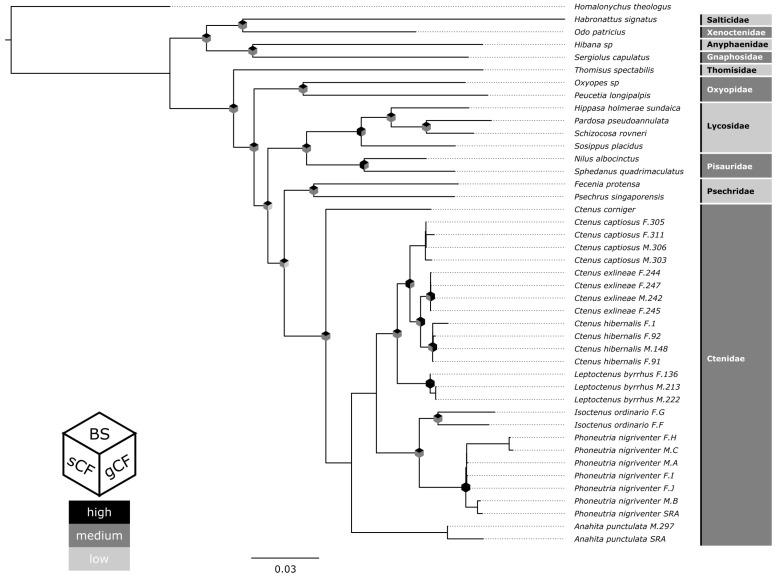
Reconstructed species-level phylogeny from concatenated matrix using IQTREE. Ultrafast bootstrap support, as well as gene and site concordance factor values are indicated by the color of the diamond placed on the inner nodes. Black, dark grey and grey indicate high, medium, and low support, respectively. Cutoff values for bootstrap: 95–100%, 90–94%, 0–90%. Cutoff values for gene concordance factors: 70–100%, 20–70%, 0–20%. Cutoff values for site concordance factors: 60–100%, 33–60%, 0–33%.

**Figure 2 toxins-15-00112-f002:**
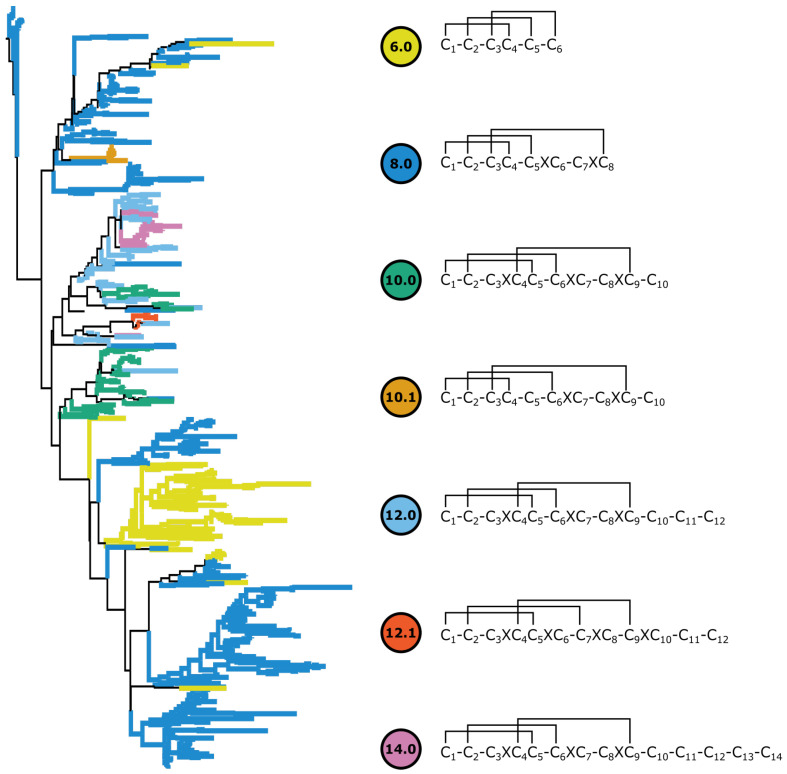
Reconstructed phylogeny of the 626 ICKs recovered from ctenids and lycosoid outgroups. Terminals are colored by their respective cysteine framework. Predicted disulfide connectivities representing all three homologous disulfide bridges shared among all ICK classes are shown to the right.

**Figure 3 toxins-15-00112-f003:**
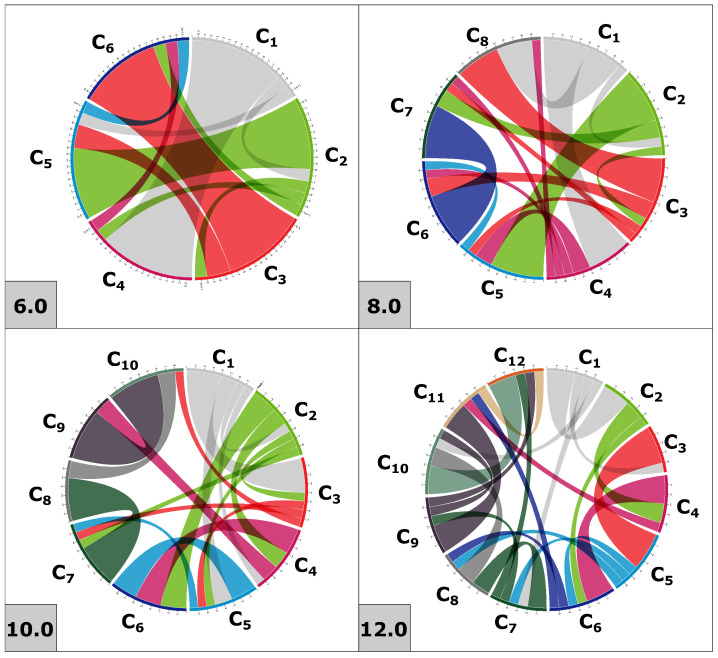
Pairwise disulfide connectivity predictions for cysteine motifs 6.0, 8.0, 10.0 and 12.0. Predictions come from a combination of four different prediction methods. Colors indicate cysteine of origin.

**Figure 4 toxins-15-00112-f004:**
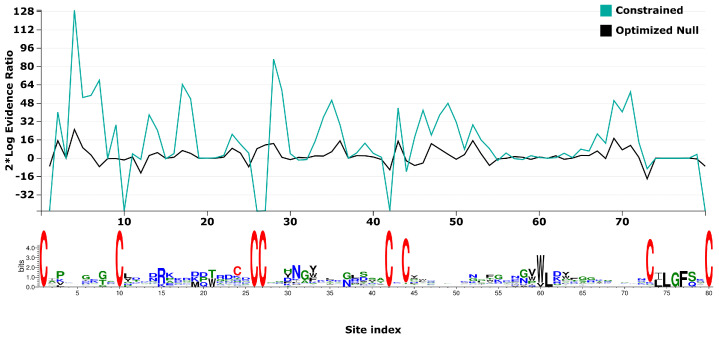
Model test statistics per site using 2*Log evidence ratio for BUSTED constrained and optimized null. Sequence logos for the alignment are presented beneath.

**Figure 5 toxins-15-00112-f005:**
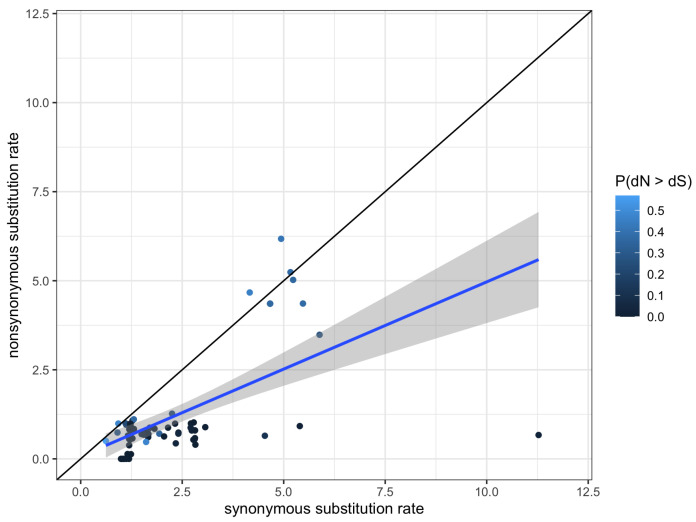
Scatter plot of synonymous substitution rate versus nonsynonymous substitution rate for each of the 80 sites of the ICK alignment. The black diagonal line indicates the null hypothesis of a lack of negative selection or positive selection where the two substitution rates are equal. Points are colored to indicate the posterior probability that a given site had evidence of pervasive positive selection. Line of best-fit is in blue, with the 95% confidence interval shaded in gray.

**Figure 6 toxins-15-00112-f006:**
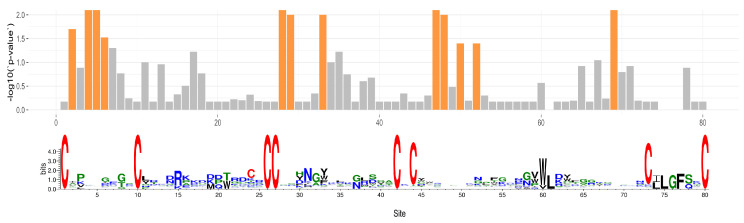
Bar plot of negative-log transformed *p*-values that a portion of branches for each site have evidence of episodic/diversifying selection. Sites with *p*-values < 0.05 are highlighted in orange. Sequence logo for the alignment presented beneath.

**Table 1 toxins-15-00112-t001:** Transcriptome assembly statistics for all ctenid samples contributed to this study, including number of SuperTranscripts and coding sequences.

Species	Sex	Sample	Transcripts	SuperTranscripts	CDS
*Anahita punctulata*	male	297	120,048	88,950	55,238
*Ctenus captiosus*	female	305	124,919	99,712	59,434
*Ctenus captiosus*	female	311	140,647	110,550	64,437
*Ctenus captiosus*	male	303	157,109	123,061	70,951
*Ctenus captiosus*	male	306	158,795	122,878	71,795
*Ctenus exlineae*	female	244	105,140	85,561	49,630
*Ctenus exlineae*	female	245	111,981	89,516	52,365
*Ctenus exlineae*	female	247	112,224	90,112	52,434
*Ctenus exlineae*	male	242	95,088	78,974	44,942
*Ctenus exlineae*	male	246	54,776	46,375	24,166
*Ctenus hibernalis*	female	91	194,576	148,947	83,512
*Ctenus hibernalis*	female	92	161,519	124,108	71,340
*Ctenus hibernalis*	male	148	202,764	157,422	81,381
*Leptoctenus byrrhus*	female	136	99,257	81,488	47,189
*Leptoctenus byrrhus*	male	213	108,687	88,723	50,313
*Leptoctenus byrrhus*	male	222	101,706	83,634	47,527

**Table 2 toxins-15-00112-t002:** Cysteine rich peptide composition in the proteomes of *C. exlineae* and *C. hibernalis*, with comparison to the expression levels from the venom gland transcriptomes of each sample per species.

Species	Sex	Sample	Peptides	ICKs	%ICK	Sum TPM	ICK TPM	%TPM
*C. exlineae*	male	242	113	5	4.42%	18,214.3	9670.5	53.1%
*C. exlineae*	female	245	127	7	5.51%	16,755.8	7002.0	41.8%
*C. exlineae*	male	246	200	13	6.50%	131,454	51,011.1	38.8%
*C. exlineae*	female	244	339	5	1.47%	42,546.4	12,227.6	28.7%
*C. exlineae*	female	247	194	5	2.58%	27,378.7	5436.3	19.9%
*C. hibernalis*	female	4926	196	8	4.08%	109,103	57,283.2	52.5%
*C. hibernalis*	female	91	525	12	2.29%	111,759	14,703.9	13.2%
*C. hibernalis*	male	148	170	7	4.12%	31,121.2	2581.5	8.3%
*C. hibernalis*	female	92	176	5	2.84%	37,128.6	2366.6	6.4%

**Table 3 toxins-15-00112-t003:** Summary of the number of peptides recovered per cysteine framework as well as the corresponding numeral indication designated by [19].

Identifier	Dinez Numeral	Motif	Total
6.0	I	$C_{1}$-$C_{2}$-$C_{3}C_{4}$-$C_{5}$-C6	123
8.0	II	$C_{1}$-$C_{2}$-$C_{3}C_{4}$-$C_{5}XC6$-$C_{7}XC_{8}$	538
10.0	V	$C_{1}$-$C_{2}$-$C_{3}XC_{4}C_{5}$-$C6XC_{7}$-$C_{8}XC_{9}$-C10	117
10.1		$C_{1}$-$C_{2}$-$C_{3}C_{4}C_{5}$-$C6XC_{7}$-$C_{8}XC_{9}$-C10	23
12.0	VI	$C_{1}$-$C_{2}$-$C_{3}XC_{4}C_{5}$-$C6XC_{7}$-$C_{8}XC_{9}$-$C_{10}$-$C_{11}$-$C_{12}$	100
12.1	VII	$C_{1}$-$C_{2}$-$C_{3}XC_{4}C_{5}XC6$-$C_{7}XC_{8}$-$C_{9}XC_{10}$-$C_{11}$-$C_{12}$	27
14.0	VIII	$C_{1}$-$C_{2}$-$C_{3}XC_{4}C_{5}$-$C6XC_{7}$-$C_{8}XC_{9}$-$C_{10}$-$C_{11}$-$C_{12}$-$C_{13}$-$C_{14}$	33

**Table 4 toxins-15-00112-t004:** Number of ICK peptides recovered for each cysteine framework per species.

Family	Species	6.0	8.0	10.0	10.1	12.0	12.1	14.0
Homalonychidae	*Homalonychus theologus*	1	4	2	0	9	0	0
Salticidae	*Habronattus signatus*	6	13	2	1	1	0	0
Xenoctenidae	*Odo patricius*	2	21	2	0	3	0	0
Anyphaenidae	*Hibana* sp.	3	10	0	1	1	0	0
Gnaphosidae	*Sergiolus capulatus*	1	11	0	1	2	0	0
Thomisidae	*Thomisus spectabilis*	2	13	3	0	2	0	1
Thomisidae	*Misumenoides formosipes*	1	4	0	1	0	0	0
Oxyopidae	*Oxyopes* sp.	0	15	9	0	2	0	0
Oxyopidae	*Peucetia longipalpis*	1	11	4	1	0	0	0
Lycosidae	*Hippasa holmerae*	1	11	5	1	3	0	0
Lycosidae	*Pardosa pseudoannulata*	0	7	1	1	0	0	0
Lycosidae	*Schizocosa rovneri*	0	10	0	0	1	0	0
Lycosidae	*Sosippus placidus*	5	26	1	1	3	0	2
Pisauridae	*Nilus albocinctus*	1	8	2	1	3	0	0
Pisauridae	*Sphedanus quadrimaculatus*	1	6	2	1	4	0	0
Pisauridae	*Pisaurina mira*	0	1	0	1	1	0	0
Pisauridae	*Dolomedes triton*	1	11	0	0	10	0	1
Psechridae	*Fecenia protensa*	1	17	4	0	2	1	1
Psechridae	*Psechrus singaporensis*	0	13	3	1	2	1	0
Ctenidae	*Ctenus corniger*	11	19	5	1	3	1	1
Ctenidae	*Anahita punctulata*	2	15	1	0	2	1	1
Ctenidae	*Ctenus captiosus*	4	10	2	1	3	1	2
Ctenidae	*Ctenus exlineae*	2	9	2	1	2	1	1
Ctenidae	*Ctenus hibernalis*	2	10	3	1	3	1	1
Ctenidae	*Isoctenus* sp.	3	11	4	0	0	1	2
Ctenidae	*Leptoctenus byrrhus*	1	12	2	1	1	1	1
Ctenidae	*Phoneutria nigriventer*	5	12	3	0	1	1	2

**Table 5 toxins-15-00112-t005:** Multiple sequence alignment schema for ICKs using the four pairs of structurally homologous cysteine residues.

Class	Loop 1	Loop 2	Loop 3	Loop 4
C6.0	$C_{1}$-$C_{2}$	$C_{2}$-$C_{3}$	$C_{4}$-$C_{5}$	$C_{5}$-$C_{6}$
C8.0	$C_{1}$-$C_{2}$	$C_{2}$-$C_{3}$	$C_{4}$-$C_{5}$	$C_{5}XC_{6}$-$C_{7}XC_{8}$
C10.0	$C_{1}$-$C_{2}$	$C_{2}$-$C_{3}XC_{4}$	$C_{5}$-$C_{6}$	$C_{6}XC_{7}$-$C_{8}XC_{9}$
C10.1	$C_{1}$-$C_{2}$	$C_{2}$-$C_{3}$	$C_{4}$-$C_{5}$-$C_{6}$	$C_{6}XC_{7}$-$C_{8}XC_{9}$
C12.0	$C_{1}$-$C_{2}$	$C_{2}$-$C_{3}XC_{4}$	$C_{5}$-$C_{6}$	$C_{6}XC_{7}$-$C_{8}XC_{9}$
C12.1	$C_{1}$-$C_{2}$	$C_{2}$-$C_{3}XC_{4}$	$C_{5}XC_{6}$-$C_{7}$	$C_{7}XC_{8}$-$C_{9}XC_{10}$
C14.0	$C_{1}$-$C_{2}$	$C_{2}$-$C_{3}XC_{4}$	$C_{5}$-$C_{6}$	$C_{6}XC_{7}$-$C_{8}XC_{9}$

**Table 6 toxins-15-00112-t006:** A statistical summary of the models fit to the ICK alignment. “Unconstrained model” refers to the BUSTED alternative model for selection, and “Constrained model” refers to the BUSTED null model for selection.

Model	log(likelihood)	Parameters	AICc	$\omega_{1}$	$\omega_{2}$	$\omega_{3}$
Unconstrained	−37,648.7	1169	77,691.4	0.06	0.09	3.35
Constrained	−37,733.9	1168	77,859.6	0.03	0.03	1.00

## Data Availability

All short read data for this project can be found using BioProject accession number PRJNA587301, accessions for reads retrieved from SRA are in the Appendix A. All other data files are in the Appendix A files. Source code for this project is available on GitHub (https://github.com/tijeco/killer_knots, accessed on 6 November 2022).

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
