# Peer review of "Killer Knots: Molecular Evolution of Inhibitor Cystine Knot Toxins in Wandering Spiders (Araneae: Ctenidae)"

_toxins, 2023, doi:10.3390/toxins15020112_

Round 1

Reviewer 1 Report

This paper reports and characterizes new transcriptomic and proteomic data of venom glands of some wandering spiders (Ctenidae) that are used to study the diversity and molecular evolution of ICK toxins. The data presented are timely and of potential interest for current and future research on spider genomics and pharmacology. Although the manuscript is well written in general and the methods employed are adequate in most cases, some revision is needed before the paper can be accepted for publication in Toxins.

The specific points that need to be addressed by the authors are:

- Lines 66-67: Please display full name of all species here (first time mentioned in the text).

- Line 76: Please add a table with detailed information for the data retrieved from NCBI, such as SRA database number or BioProject number for each species.

- Line 79: This temperature range appears to be in Fahrenheit (not Celsius) degrees... please correct.

- Line 156: Please indicate which specific algorithm was employed in TRIMAL.

- Lines 232-279: This entire section is not very clearly written. I recommend rewording.

- Table 1: Maybe add an additional column with the number of longest isoform in each case (as a proxy of possible number of distinct/unique genes).

- Figure 1: Please check the cut-off values employed in all cases. For example, the ultrafast bootstrap conducted in IQ-TREE has cut-off values different from the traditional bootstrap (>95% vs. >70%) (see IQTREE manual for details: http://www.iqtree.org/doc/Frequently-Asked-Questions#how-do-i-interpret-ultrafast-bootstrap-ufboot-support-values ). This is important.

- Table 3 (head of last column): Please change ‘total’ to ‘length’.

- Figure 2: Please explain the meaning of colors in the caption.

- Lines 335-338: This sentence is unclear, please reword.

- Line 362: Please remove extra ‘only’.

- Line 389: Please change ‘the same’ to ‘similar’ (or ‘comparable’).

- Lines 392-395: This appears fairly speculative...

- Lines 401-402: Please add reference(s) for this statement.

- There are some minor language and grammar issues that need correction: use of commas, use of that/which, the English is sometimes unnatural. The paper probably would benefit of a thorough proof-reading by a native English speaker before resubmission.

I hope the authors find my comments helpful.

Author Response

We thank the reviewer for their helpful comments. Please see our point-by-point responses below:

  • Lines 66-67: Please display full name of all species here (first time mentioned in the text).).
    • This has been addressed
  • Line 76: Please add a table with detailed information for the data retrieved from NCBI, such as SRA database number or BioProject number for each species.
    • The supplemental file has been included
  • Line 79: This temperature range appears to be in Fahrenheit (not Celsius) degrees... please correct.
    • This has been corrected
  • Line 156: Please indicate which specific algorithm was employed in TRIMAL.
    • This has been updated.
  • Lines 232-279: This entire section is not very clearly written. I recommend rewording.
    • This has been reworded
  • Table 1: Maybe add an additional column with the number of longest isoform in each case (as a proxy of possible number of distinct/unique genes).
    • Supertranscripts are included in that table, which serve as a proxy for distinct / unique genes.
  • Figure 1: Please check the cut-off values employed in all cases. For example, the ultrafast bootstrap conducted in IQ-TREE has cut-off values different from the traditional bootstrap (>95% vs. >70%) (see IQTREE manual for details: 
    • All ultrafast bootstrap values were at the 100 mark, except for the psechrid+ctenid clade, which was 83.
  • Table 3 (head of last column): Please change ‘total’ to ‘length’.
    • This has been adjusted
  • Figure 2: Please explain the meaning of colors in the caption.
    • The explanation for the colors has been added.
  • Lines 335-338: This sentence is unclear, please reword.
    • This has been reworded.
  • Line 362: Please remove extra ‘only’.
    • This has been removed.
  • Line 389: Please change ‘the same’ to ‘similar’ (or ‘comparable’).
    • This has been changed.
  • Lines 392-395: This appears fairly speculative…
    • The fairly speculative portions have been removed.
  • Lines 401-402: Please add reference(s) for this statement.
    • References have been added.
  • There are some minor language and grammar issues that need correction: use of commas, use of that/which, the English is sometimes unnatural. The paper probably would benefit of a thorough proof-reading by a native English speaker before resubmission.
    • The authors are native English speakers. A thorough proof-reading was done.

Reviewer 2 Report

It is interesting to consider that ICK, the most widely found C8.0 cysteine framework, was the origin, from which one cysteine pair was lost to generate the C6.0 framework. On the other hand, it is wondering how the gene that increased the number of cysteines by around 2-fold was obtained.

Four programs predict combinations of disulfide bonds, but it is awaiting corroboration by empirical structural analysis.

It is also interesting to identify amino acid residues that are predicted to coevolve.

It is desirable to investigate the relationship between the chemical properties of such amino acids and the physiological activity of ICKs containing them.

Author Response

We thank the reviewer for their helpful comments. Please see our point-by-point responses below:

  • It is interesting to consider that ICK, the most widely found C8.0 cysteine framework, was the origin, from which one cysteine pair was lost to generate the C6.0 framework. On the other hand, it is wondering how the gene that increased the number of cysteines by around 2-fold was obtained.
    • We agree that this question is quite interesting, but addressing this is unfortunately beyond the scope of this experiment and dataset.
  • Four programs predict combinations of disulfide bonds, but it is awaiting corroboration by empirical structural analysis.
    • Empirical studies confirming the 3D structure of these proteins would be a great follow-up study the one presented here.
  • It is also interesting to identify amino acid residues that are predicted to coevolve.
    • We ran these analyses and did find interesting results that were presented in the original submission.
  • It is desirable to investigate the relationship between the chemical properties of such amino acids and the physiological activity of ICKs containing them.
    • We agree, but this would require a great deal of additional work. The presented work set the foundation for many future studies.

Reviewer 3 Report

This manuscript exams the evolution relatedness of ICK’s motif in wandering spiders including addition of venomic data of animals, bioinformatic analysis was subsequently undertaken on this data. The authors state the key contribution is novel conformation however I seem to have missed it altogether in the manuscript. To my mind the novel contribution is a very large data set of venomic data to conduct bioinformatic analysis with. My recommendation is major revision.

There is a large contribution of venomic data for species in this research however I am not going to lie – I feel like I have been hit by a bioinformatic bulldozer and got to the end of the manuscript confused and not sure what the point was in the end. There is an astounding amount of bioinformatics gone into this study, but I feel it has not been put into a biological context for the reader nor is the analysis well supported.

Details are missing (experimental and bioinformatically), and clarity makes the research lost or renders the reader unsure of the authors key assessment and the authors knowledge of the wandering spider ICK toxins in a structural and physiological context.

A key concern regarding this manuscript is that authors appear to believe a cysteine framework of 6 cysteines automatically means it will form an ICK structure. Additionally, something having an ICK sequence similarity mean that the sequence is functionally active, thus overinflating the number putative ICK’s found in databases that at the end of the day mat not be ICK’s. The phylogenies presented here do not meet the bioinformatic minimum standards for well supported phylogeny which should be the key focus of the research.

The authors state that the key contribution of the study is a that novel conformation was found. There is no mention of this novel conformation in the discussion or the importance of this. Nor is there a structural study undertaken to support this ‘predicted’ novel confirmation. At the end of the day anything done bioinformatically/computationally is only a prediction until ground-truthing is completed and the sequence in question resolved as a structure with a function.  In fact, on L378 the authors state that the connectivities servers are imprecise and yet this forms the basis of the manuscript.

The study is very focus only on the Cys and connectivities, there is acknowledgement that additional residues are important for proteins in binding etc as the ICK has a very specific disulphide formation and fold, there will be residues important for enabling such a structure to form and it appears these have not been considered in the analysis.

Perhaps some of these concerns may be addressed in the below specific comments.

Comments and formatting corrections

When referencing the describing author of a species the reference needs should be a font size smaller than the normal text to indicate it is the author of the species e.g., L48 but also it should be used in the first instance of the use of the name in this case L39.

Lack of reference Paragraph (L49-58) no reference to the initial ICK cysteine framework research and this needs to be corrected examples references below.

Narasimhan, L., Singh, J., Humblet, C., Guruprasad, K. and Blundell, T. (1994) Snail and spider toxins share a similar tertiary structure and `cystine motif'. Nat. Struct. Biol. 1, 850±852.

Pallaghy, P. K., Nielsen, K. J., Craik, D. J. and Norton, R. S. (1994) A common structural motif incorporating a cystine knot and triple-stranded b-sheet in toxic and inhibitory polypeptides. Protein Sci. 3, 1833± 1839

L 52 – wording is confusing are you saying the 98 cysteine rich toxins were discovered in the original research or in the recent research from P. nigriventer transcript-/proteome?

L65 Ctenidae should be stated to be the phyla level – in this instance family and the first species genus needs to be written out in full e.g. Ctenus hibernalis, C. exlineae……. This should also incorporate all the describing species authors in the first instance, as in L48 and references included in reference listing.

State who taxonomically identified the animals in the methods and materials and state who the Brazilian collaborate was in the text, the initials if it is an included author.

L79 72-77 degrees C? Is this a typo and you mean Fahrenheit? 77 degrees Celsius seems beyond the upper limit of heat tolerance in spiders as far as I know.

Method clarifications: How many of each individual species was milked? Was the venom pooled for proteomics/transcriptomic analysis if more than one specimen of each species? Table 1 infers each specimen had an individual transcriptome but was the venom pooled or kept as individuals for analysis.

How was the venom gland processed RNA extraction? – whole section missing in methods.

 L112 cross contamination producing large number of transcripts – were the samples not barcoded and how many samples were run in each lane for sequencing?

How many extractions were sent for sequencing in total?

L 122 – Just because a protein isn’t expressed highly in RNA it doesn’t mean it is not functionally relevant – it just means that it wasn’t highly expressed at that point in time material for RNA analysis was collected and should not be immediately disregarded.

L134 poorly worded grammar “singly charge analytes” – amend

L150 reference the downloaded RNA seq reads, even in supp data

L143 The version of BUSCO used is a little old – The analysis should be quick to rerun in the new version if any variance in conserved gene and assembly of transcriptomes.

L149 Elucidate what do you consider to be a complete coding sequence

Figure 1 – I would question the division cut-offs of high, medium and low for the node supports.  nodes would only be accepted as >80% as being a well-supported. The phylogeny here is seemingly quite unsupported with all clades falling into medium and low and absolutely none in the accepted range of 80% making this an unsupported phylogeny.

L170 what e-value cut-off was used for the HMMM?

L 307 I question the parameters that are presented here for a full sequence to have 6 Cys minimum and a signal peptide – does this mean the phylogeny is full of partial sequences i.e., no stop codon?

L 308 were the sequences used from the KNOTTIN database all been assayed for their functionality or are some of these sequences putative ICK’s from other bioinformatic studies?

Table 3 – I find the headings a little lacking and not clear for the reader at first glance is the identifier I am assuming is based on the number of cysteines and if they have and additional residue, they are called 0. 1? excuse my ignorance but what is a Dinez numeral? and is the total the number of peptides found with that motif?

L 320 state the author as Somebody et al., and just use the numerical reference number

Table 4 “sp” should not be italicised in the table and has a full stop afterwards. e.g. Oxyopes sp.

Figure 2. What do the colours mean in the predictions? Pairwise predictions with what sequence?

Figure 3 – This needs work – the basal clade goes off into nothingness at the top of the page and or the end of the node is obscured by colour overlaid. I am assuming this is the outgroup and I really shouldn’t have to assume – tell the reader – overlay it with the family/genus is it is – is the meant be overlaid on the previous phylogeny? Again, I am assuming – you still need to put branch lengths and node support values on the branches. And if this is overlaid with the previous phylogeny which has little support it doesn’t really mean very much. Perhaps I have misunderstood the entire Figure in which case it needs better presentation

L 367 The fifth amino acid – this needs more detail – you are describing the results, and this means nothing to the reader – is it a particular residue? Is it the fifth amino acid from the N terminal?

I find the discussion really lacking – I do not get a feel for a well-supported phylogeny of the ICK framework/structure evolution in wandering spiders nor how this fits in with venom research. Perhaps there is a different type of analysis can be conducted to examine these sequences as it stands there is not clear outcome of the evolution from the analysis conducted here.

Author Response

We thank the reviewer for their help comments. Please see our point-by-point responses below:

  • When referencing the describing author of a species the reference needs should be a font size smaller than the normal text to indicate it is the author of the species e.g., L48 but also it should be used in the first instance of the use of the name in this case L39.
    • We are unaware of this in the journal’s style guide, so we did not change the font size of the text.
  • Lack of reference Paragraph (L49-58) no reference to the initial ICK cysteine framework research and this needs to be corrected examples references below.
    • References have been added
  • L 52 – wording is confusing are you saying the 98 cysteine rich toxins were discovered in the original research or in the recent research from P. nigriventer transcript-/proteome?
    • This has been rephrased for clarity.
  • L65 Ctenidae should be stated to be the phyla level – in this instance family and the first species genus needs to be written out in full e.g.  Ctenus hibernalis, C. exlineae……. This should also incorporate all the describing species authors in the first instance, as in L48 and references included in reference listing.
    • We are unaware of this being common practice, so we left it as is
  • State who taxonomically identified the animals in the methods and materials and state who the Brazilian collaborate was in the text, the initials if it is an included author.
    • This has been updated
  • L79 72-77 degrees C? Is this a typo and you mean Fahrenheit? 77 degrees Celsius seems beyond the upper limit of heat tolerance in spiders as far as I know.
    • This has been updated.
  • Method clarifications: How many of each individual species was milked? Was the venom pooled for proteomics/transcriptomic analysis if more than one specimen of each species? Table 1 infers each specimen had an individual transcriptome but was the venom pooled or kept as individuals for analysis.
    • The methods has been updated for clarity.
  • How was the venom gland processed RNA extraction? – whole section missing in methods.
    • Lines 93-96 goes into those details
  • L112 cross contamination producing large number of transcripts – were the samples not barcoded and how many samples were run in each lane for sequencing?
    • The citations directly address that cross-contamination is an issue regardless of barcoding.
  • How many extractions were sent for sequencing in total?
    • This is in table 1
  • L 122 – Just because a protein isn’t expressed highly in RNA it doesn’t mean it is not functionally relevant – it just means that it wasn’t highly expressed at that point in time material for RNA analysis was collected and should not be immediately disregarded.
    • Making use of lowly expressed transcripts is not within the scope of this investigation.
  • L134 poorly worded grammar “singly charge analytes” – amend
    • This has been updated
  • L150 reference the downloaded RNA seq reads, even in supp data
    • This has been updated.
  • L143 The version of BUSCO used is a little old – The analysis should be quick to rerun in the new version if any variance in conserved gene and assembly of transcriptomes.
    • Rerunning the etire upstream process of this analysis with a slightly modified version of the software is not within the scope of this investigation. The orthologs used in the database are well established and don't drastically change from version to version.
  • L149 Elucidate what do you consider to be a complete coding sequence
    • We have rephrased for clarity.
  • Figure 1 – I would question the division cut-offs of high, medium and low for the node supports.  nodes would only be accepted as >80% as being a well-supported. The phylogeny here is seemingly quite unsupported with all clades falling into medium and low and absolutely none in the accepted range of 80% making this an unsupported phylogeny.
    • Concordance factors don’t always have as high of support values as bootstrap values, the bootstrap values were very high, so the support isn’t in question.
  • L170 what e-value cut-off was used for the HMMM?
    • This has been updated.
  • L 307 I question the parameters that are presented here for a full sequence to have 6 Cys minimum and a signal peptide – does this mean the phylogeny is full of partial sequences i.e., no stop codon?
    • We painstakingly made sure no partial sequences were included.The phrasing has been updated for clarity.
  • L 308 were the sequences used from the KNOTTIN database all been assayed for their functionality or are some of these sequences putative ICK’s from other bioinformatic studies?
    • This is an expert curated dataset, so our current best guess of the truth. We have made adjustments for clarity.
  • Table 3 – I find the headings a little lacking and not clear for the reader at first glance is the identifier I am assuming is based on the number of cysteines and if they have and additional residue, they are called 0. 1? excuse my ignorance but what is a Dinez numeral? and is the total the number of peptides found with that motif?
    • Table 3 didn’t really serve a purpose, it has been removed.
  • L 320 state the author as Somebody et al., and just use the numerical reference number
    • This has been updated.
  • Table 4 “sp” should not be italicised in the table and has a full stop afterwards. e.g. Oxyopes sp.
    • This has been updated.
  • Figure 2. What do the colours mean in the predictions? Pairwise predictions with what sequence?
    • The caption has been updated to clarify what the cors in
  • Figure 3 – This needs work – the basal clade goes off into nothingness at the top of the page and or the end of the node is obscured by colour overlaid. I am assuming this is the outgroup and I really shouldn’t have to assume – tell the reader – overlay it with the family/genus is it is – is the meant be overlaid on the previous phylogeny? Again, I am assuming – you still need to put branch lengths and node support values on the branches. And if this is overlaid with the previous phylogeny which has little support it doesn’t really mean very much. Perhaps I have misunderstood the entire Figure in which case it needs better presentation
    • This is simply meant to represent how the different cysteine topologies identified group together. 
  • L 367 The fifth amino acid – this needs more detail – you are describing the results, and this means nothing to the reader – is it a particular residue? Is it the fifth amino acid from the N terminal?
    • This has been updated for clarity
  • I find the discussion really lacking – I do not get a feel for a well-supported phylogeny of the ICK framework/structure evolution in wandering spiders nor how this fits in with venom research. Perhaps there is a different type of analysis can be conducted to examine these sequences as it stands there is not clear outcome of the evolution from the analysis conducted here.
    • The discussion has been updated to make it less lacking.